# Pyrolysis Kinetics of Lignin-Based Flame Retardants Containing MOFs Structure for Epoxy Resins

**DOI:** 10.3390/molecules28062699

**Published:** 2023-03-16

**Authors:** Tianyu Yao, Ruohan Yang, Cong Sun, Yuzhu Lin, Ruoqi Liu, Hongyu Yang, Jiajia Chen, Xiaoli Gu

**Affiliations:** Jiangsu Co-Innovation Center of Efficient Processing and Utilization of Forest Resources, Jiangsu Provincial Key Lab for the Chemistry and Utilization of Agro-Forest Biomass, College of Chemical Engineering, Nanjing Forestry University, Nanjing 210037, China

**Keywords:** lignin, epoxy resin, flame retardant, thermal stability, kinetics

## Abstract

This study describes the preparation of a lignin-based expandable flame retardant (Lignin-N-DOPO) using grafting melamine and covering 9,10-dihydro-9-oxa-10-phosphaphenanthrene-10-oxide (DOPO) using the Mannich reaction. Then, through in situ growth, a metal-organic framework (MOF) HKUST-1 (e.g., Cu_3_(BTC)_2_, BTC = benzene-1,3,5-tricarboxylate)/lignin-based expandable flame retardant (F-lignin@HKUST-1) was created. Before that, lignin epoxy resin containing phosphorus (P) and nitrogen (N) components had been created by combining epoxy resin (EP) with F-lignin@HKUST-1. Thermogravimetric analysis was used to examine the thermal characteristics of epoxy resin (EP) composite. The findings indicate that the thermal stability of EP is significantly affected by the presence of F-lignin@HKUST-1. Last but not least, the activation energy (E) of EP/15% F-lignin@HKUST-1 was examined using four different techniques, including the Kissinger-SY iteration method, the Ozawa-SY iteration method, the Lee-Beck approximation-iteration method, and the Gorbatchev approximation-iteration method. It was discovered that the activation energy was significantly higher than that of lignin. Higher activation energy suggests that F-lignin@HKUST-1 pyrolysis requires more energy from the environment, which will be significant about the application of lignin-based flame retardants.

## 1. Introduction

Epoxy resin (EP), as one of the important thermosetting plastics, has high mechanical strength, excellent adhesion, chemical resistance, and electrical insulation, etc., and has been widely used in construction, automotive, electronics, and aerospace fields [1,2,3]. However, the inherent flammability and smoke emission of EP greatly limit its application in some specific areas [4,5,6]. Therefore, the flame-retardant modification of EP is of great importance.

Materials made of biomass offer various benefits, including being renewable [7], relatively non-polluting [8], widely available, and carbon neutral. Animals, plants, and microbes make up biomass, which is employed in a variety of industries such as the pharmaceutical [9], industrial [10], and packaging sectors [11]. Because biomass produces a lot of carbon after burning, lignin has been employed in the field of flame retardants as a carbon-forming agent [12,13,14,15,16].

Mandlekar et al. [17] investigated the patterns of thermal stability and flame retardancy of bio-based polyamide 11 (PA) having different lignin contents. Microcomposites based on PA and containing 5, 10, 15, and 20 wt.% lignin were prepared using a twin-screw extruder. Morphological analysis showed that the resulting microcomposites had good interfacial interactions and uniform distribution of lignin particles. In addition, thermogravimetric analysis in an inert atmosphere showed that sulfated lignin was able to produce lower coke residues (41–48 wt.% at 600 °C), and unlike sulfated lignin, sulfonated lignin was able to provide higher thermal stability as well as higher coke residues (55–58 wt.%). In addition, vertical flame-spread tests clearly showed that 15 wt.% was the optimum value for kraft or sulfonated lignin loading to achieve improved flame retardancy and V1 rating. In addition, cone-calorimetric tests were used to investigate the forced combustion behavior; in particular, the microcomposites containing sulfonated lignin showed significantly lower peak heat-release rates (−51%) and total heat release (−23%), lower flat-mass loss rates, and a significant mass increase in the final residue (~9 wt.%). In contrast, microcomposites containing kraft lignin showed the opposite effect, as the heat release rate (HRR) and total heat release (THR) values increased in the presence of kraft lignin.

Despite the fact that lignin may be applied directly to materials to act as a flame retardant, the result frequently falls short of expectations, hence it is typically modified chemically. Due to their abundance and strong reactivity, the hydroxyl groups (phenolic or aliphatic hydroxyl groups) in the lignin structure permit chemical change. The use of PN-Lignin as a flame retardant for many polymeric materials, including polypropylene (PP), polylactic acid (PLA), EP, and polyurethane (PU) materials, is frequently investigated. In flame-retardant systems, elemental nitrogen serves as a gas source and expands the material. In flame retardants, elemental phosphorus serves as an acid source and can also be employed as a dehydrating agent, most frequently in the form of POCI_3_, polyphosphoric acid, DOPO (9,10-dihydro-9-oxa-10-phosphaphenanthrene-10-oxide), and its derivatives. These phosphorus-containing compounds can generate acids that, when heated to a high temperature, dehydrate the carbon layer and esterify polyols, causing the carbon source to create a phosphorus-rich charred layer.

The flame-retardants effect of lignin modified by phosphorus and nitrogen has been improved, but its flame-retardant efficiency and smoke-suppression performance sometimes cannot meet the demand, for example not effectively reducing the release of CO and the emission of carbon particles, which will still have a considerable impact on the environment. According to the research in recent years, metal-organic frameworks(MOFs) materials have good adsorption effect on smoke generation and good catalytic effect on CO oxidation due to their structural characteristics. Zhang et al. [18] used MOFs as precursors to link two different LDHs into composite nanomaterials, and the specimens passed the V-0 rating in the UL-94 test. Xu et al. [19] prepared functionalized reduced graphene oxide (RGO) with Co-ZIF (zeolite imidazolium framework-67) adsorbed borate ions (ZIF-67/RGO-B). The peak heat-release rate (PHRR), total heat release rate (THR), and maximum smoke concentration (Ds, max) of the composites with ZIF-67/RGO-B doping of 2 wt.% were reduced by 65.1%, 41.1% and 66.0%, respectively, compared to the pure EP. Hou et al. [20] successfully synthesized iron- and cobalt-based MOFs and incorporated them into polystyrene (PS) as flame retardants. Thermogravimetric and cone-calorimetric analyses showed that the MOFs had good flame-retardant effects. Hou et al. [21] synthesized a Co-based metal organic backbone (P-MOF) with a phosphorus-based structure using a facile hydrothermal reaction and added the P-MOF to an epoxy resin (EP) to enhance its flame retardancy. Results from the conical calorimeter and steady-state tube-furnace tests showed a significant reduction in total flue-gas yield and total CO.

The purpose of this study was to develop an EP flame retardant that is synthetically simple, utilizing alkaline lignin, melamine, DOPO, and HKUST-1 (a metal-organic framework (MOF), Cu_3_(BTC)_2_, BTC = benzene-1,3,5-tricarboxylate). To create lignin-based flame retardants, alkaline lignin was altered in this study to enhance the reaction sites. Then, a phosphor-nitrogen lignin-based flame retardant was created by reacting the phenolized modified lignin with DOPO and melamine complex (Lignin-N-DOPO), and then HKUST-1/lignin-based flame retardant (F-lignin@HKUST-1) was prepared by introducing the MOFs structure as its basis. The flame-retardant properties were significantly enhanced by the addition of EP, and with the use of the Kissinger-SY iterative approach, the Ozawa-SY iterative method, the Lee-Beck approximation, and the Gorbatchev approximation, the kinetics of its pyrolysis were examined.

## 2. Results

### 2.1. Thermogravimetric Analysis

The thermal-stability and thermal-degradation processes of EP and its composites involved are investigated using thermodynamic analysis curves (from room temperature to 700 °C). The TG and DTG curves of EP and its composites are depicted in Figure 1, and Table 1 contains information on the starting thermal decomposition temperature (T_d5%_), the maximum thermal decomposition rate temperature (T_dmax_), and the quantity of residual carbon. Figure 1a illustrates the one-step thermal deterioration of the pure EP sample in N_2_. The decomposition begins at 333.7 °C and reaches its maximum rate at 405.3 °C, leaving only 16.21 wt.% of carbon behind. The thermogravimetric curves of EP/15% F-Lignin@HKUST-1 are similar to those of pure EP, but exhibit different thermal stability.

Figure 1 shows that the EP/15%F-lignin@HKUST-1 started to degrade before the EP, as may be inferred. T_dmax_ showed a trend toward decrease, as shown in Table 1, and this finding can be related to the early phosphorus breakdown, the quick DOPO breakdown, and the catalytic action of the breakdown products of MOFs, which makes it easier to create fast-catalytic carbon residues. Additionally, it can be seen from Table 1 that the degradation temperature range of EP/F-lignin@HKUST-1 is smaller, indicating that it can complete the decomposition more quickly and act as a catalyst and carbon promoter [22]. These results show that the addition of a MOFs structure can improve the residual carbon amount of the composite as shown by the higher residual carbon amount at 700 °C compared to pure epoxy resin. These findings suggest that the synergistic interaction between DOPO and MOFs can facilitate the carbon-production process in thermoset composites based on epoxy.

### 2.2. Kinetic Analysis

The apparent activation energy at the appropriate conversion is frequently determined by linear-regression analysis utilizing the iso-conversion method using the pyrolysis curves produced by the pyrolysis process of solid-state reactants. The most crucial step is picking an appropriate approximation for the temperature integral. With different conversions, the Lee-Beck-approximation and Gorbatchev-approximation iterative methods are employed for *P*(*u*), while the third-order expression of Senum-Yang is selected for the temperature-integral function *P*(*u*) and iterated by the Kissinger-SY and Ozawa-SY iterative methods. The apparent activation energy Eα at different conversions can be obtained by iteratively calculating the pyrolysis conversion curves of samples Al-lignin, EP/15% F-Lignin@HKUST-1 in 0.05 steps from 0.15–0.90, etc. Based on the aforementioned four methods of calculation, Table 2 and Table 3, respectively, present the apparent activation energy *Eα* and R^2^ (regression coefficient) at various conversions.

Table 2 and Table 3 show that the regression coefficients R^2^ for the apparent activation energy of the Al-lignin pyrolysis process at various conversions. These values may be impacted by experimental errors and result in a mediocre fit, whereas the R^2^ of the regression coefficients at different conversions. With the increased conversion (0.20 < α < 0.80), the regression coefficients’ R^2^ values ranged from 0.95 to 0.99. This indicates that the experimental data of sample F-Lignin@HKUST-1 is good on the one hand, while on the other hand the pyrolysis process of this sample is probably a single-step reaction kinetic. The apparent activation-energy values acquired by the four iterative approaches are quite near to each other, and the fit of the data obtained from the simultaneous analysis of the three distinct materials using the four iterative methods is high, suggesting that the findings computed by the four iterative methods are more accurate. Linear-regression analysis plots of iso-conversions of samples Al-lignin and F-Lignin@HKUST-1 are shown below in Figure 2 and Figure 3, respectively.

Table 2 and Table 3 show that the apparent activation energies Eα obtained for the two samples using the four different calculation methods are nearly identical. One of the four calculation methods can be used to determine the apparent activation-energy (Eα) value of lignin, which can then be used to study the activation-energy law during lignin pyrolysis. According to the Ozawa-iterative kinetic model, Figure 4 depicts the curve of conversion versus activation energy for the samples Al-lignin and EP/15% F-Lignin@HKUST-1, respectively. The energy barrier needed for chemical reactions to take place is known as the activation energy; the higher the activation energy, the more challenging the reaction. It establishes the reaction rate’s responsiveness and sensitivity [23]. As a result, the various activation energies at various conversions highlight both the complexity of the reactions and the multi-stage nature of the thermal degradation of solid materials. It also demonstrates the intricacy of the two materials’ pyrolysis and chemical changes, which should have involved many reactions at various phases.

Figure 4 shows that the activation energy needed for the pyrolysis of the sample Al-lignin is low at the early stages of decomposition (0.05 < α < 0.10), which is because the initial process primarily involves the evaporation of water as well as some small molecules, and the energy needed for the evaporation of water is typically low. The local extreme points of EP/15% F-lignin@HKUST-1’s energy versus conversion curve demonstrate that its water content was lower, and its water-evaporation phases were shorter, than in the aforementioned TG and DTG studies; all of them were roughly in the α < 0.10 interval. Similar changes in activation energy were seen in the sample Al-lignin during the initial stages of decomposition.

Both samples begin to reach the severe-pyrolysis phase, which breaks numerous chemical bonds and demands more energy. Lignin pyrolysis is a heat-absorbing process. Al-lignin, one of the samples, had a quick apparent activation energy in the interval of conversion 0.1 < α < 0.4. It eventually rose from around 20 kJ/mol to a range of 100–120 kJ/mol and stayed steady. The apparent activation energy of sample EP/15% F-lignin@HKUST-1 increased rapidly from about 70 kJ/mol to about 130 kJ/mol in the range of conversion 0 < α < 0.1 and remained stable.

The value of the sample Al-lignin activation energy fluctuated slightly as the conversion increased (0.50 < α < 0.70), and the reaction-activation energy showed a downward trend. This was explained by the fact that the formal pyrolysis process of lignin reached its final stage and the chemical bond-breaking process in the lignin molecule slowed down. The activation energy needed for lignin pyrolysis again exhibits a steep rising trend as the conversion rises (0.70 < α < 0.80). This is because the inorganic salts in the lignin molecule start to break down, which consumes a lot of energy. The inorganic salt degradation process was reaching its conclusion when the conversion kept rising (0.80 < α < 0.90), and as a result, the apparent activation energy of lignin began to decline.

The conversion for the sample EP/15% F-lignin@HKUST-1, however, very slightly increased over the range of 0.15 to 0.85 and stayed constant at 130 kJ/mol. The inclusion of flame-retardant components to the modified materials raised the energy barrier during the pyrolysis of lignin, resulting in greater activation energies for the main pyrolysis phases of those materials than for Al-lignin. Also, the difference of the added flame retardant resulted in the activation energy of the main pyrolysis stage of sample EP/15% Lignin-N-DOPO being greater than that of the main pyrolysis stage of sample EP/15% F-lignin@HKUST-1. It can be shown that sample EP/15% F-lignin@HKUST-1 had a shorter main pyrolysis phase, a bigger pyrolysis rate than the former, and a higher fluctuation in conversion when combined with the TG and DTG assays on the modified lignin samples above.

The modified lignin samples’ activation energy versus conversion curves showed some degree of stability throughout a range of conversions, suggesting that the pyrolysis process may be adequately captured by a single-step reaction kinetic-mechanism equation. A multi-step reaction kinetic equation could be required since the activation energy of Al-lignin was increasingly complex with the conversion, and since the pyrolysis process could not be well characterized by a single-step reaction kinetic equation.

### 2.3. Mechanic Equations

The phases of conversion α 0.1 to 0.9 were replaced into the universal integral Equation (1) and differential Equation (2), respectively, in four sets of lignin pyrolysis data with varying heating rates, and the Avrami-Erofeev equation was chosen as the mechanism function (AE_n_).
(1)$$\ln\left[\frac{G\left(\alpha \right)}{T-T_{0}}\right]=\ln\frac{A}{\beta }-\frac{E}{RT}$$

Avrami-Erofeev equation (AE_n_):(2)$$\ln\left\{\frac{\frac{d\alpha }{dT}}{f\left(\alpha \right)\left[\frac{E\left(T-T_{0}\right)}{RT^{2}}+1\right]}\right\}=\ln\frac{A}{\beta }-\frac{E}{RT}$$

The lignin pyrolysis data were calculated using the mechanism functions as equations: AE_4_, AE_5_, AE_6_.

The universal integral equation and differential equation were used for the four sets of EP/15% F-lignin@HKUST-1 pyrolysis data with various heating rates. The mechanism functions were chosen as equations AE_2_, AE_3_, and AE_4_, respectively, and the apparent activation energy produced from the four iterative calculation techniques was utilized to compare and filter the levels of the mechanism functions. The output of the computation is displayed in Table 4 and Table 5 below.

The calculation results in Table 4 and Table 5 demonstrate that the kinetic parameters acquired by the differential-equation approach and the universal-integral method are quite near to one another, demonstrating the relative reliability of the findings produced by the two methods. Table 4 shows the apparent pyrolysis activation energies of Al-lignin, which vary from 65 to 76 kJ/mol, 84 to 98 kJ/mol, and 103 to 120 kJ/mol, respectively. R^2^, the fitting coefficient, is also more than 0.95. The average apparent activation energy Eα of lignin pyrolysis, as shown in Table 2, is 99.82 kJ/mol, making AE_6_ the most likely candidate for the mechanism function. Table 5 Mechanism functions AE_2_, AE_3_ and AE_4_ are respectively applied. The apparent activation energies of EP/15% F-lignin@HKUST-1 pyrolysis of samples are 73–113 kJ/mol, 115–175 kJ/mol and 158–237 kJ/mol, respectively, and the fitting coefficients R^2^ are all higher than 0.9. The apparent activation energy of pyrolysis of the sample obtained in Table 3 above is mostly between 128 kJ/mol and 145 kJ/mol, so the most likely mechanism function is determined as AE_3_.

The most probable pyrolysis-mechanism function of Al-lignin is AE6, and that of EP/15% F-lignin@HKUST-1 is AE_3_ according to the universal integral and differential equation methods.

### 2.4. Compensation Effect

An essential component of the research of thermal-analysis kinetics is the kinetic compensatory effect. By using the equal-conversion approach, the appropriate pre-exponential factor *lnA* may be determined once the mechanism function of the pyrolysis process of solid reactants has been established. In Table 6 and Table 7, the apparent activation energy Eα and preexponential factor *lnA* of the lignin samples EP/15% F-lignin@HKUST-1 and Al-lignin are shown, respectively, based on the Kissinger-SY iteration approach and the Ozawa-SY iteration methods, where the mechanism function is the most plausible mechanism function corresponding to it.

When the mechanism function is the most likely, Figure 5 and Figure 6 illustrate the related compensating-effect curves for lignin Al-lignin and sample EP/15%F-lignin@HKUST-1 based on Kissinger-SY iteration technique and Ozawa-SY iteration method, respectively. Figure 5 shows that in the pyrolysis of the three materials, the apparent activation energies Eα and *lnA* exhibit some linear connection.

Figure 5 illustrates this relationship between the apparent activation energy, E, during lignin pyrolysis and the preexponential component *lnA*. This relationship occurs under the assumption that mechanism function AE6, *lnA* = 0.1589*E* − 1.9599, with an R^2^ of 0.88, is the equation that fits using the Kissinger-SY technique. The fitted equation using the iterative Ozawa-SY approach is *lnA* = 0.1589*E* − 1.9595 with an R^2^ of 0.88. According to the Kissinger-SY iteration approach, the fitted equation for sample EP/15%F-lignin@HKUST-1 is *lnA* = 0.3574*E* − 25.0022, with R^2^ equal to 0.88. According to the Ozawa-SY iteration method, the fitted equation is *lnA* = 0.3574*E* − 25.0021, with R^2^ equal to 0.88. The two samples’ calculation outcomes using the two approaches are almost identical, demonstrating the validity of the calculation outcomes. All of the R^2^ coefficients for linear fitting were larger than 0.88, although there was no evidence of substantial linear association. This suggests that the two samples’ pyrolysis processes are rather complicated, and it is challenging to characterize the reaction-mechanism function of pyrolysis by a mechanism function with matching physical relevance. This is due to the fact that several various chemical processes take place during the pyrolysis of materials, and the power series of the mechanism function may be rather complicated.

## 3. Experiments

### 3.1. Materials

Alkaline lignin (Al-lignin) was purchased from TCI Shanghai Co., Ltd., Shanghai, China. DOPO, carboxymethyl cellulose sodium (CMC) and 4,4-diaminodiphenylmethane (DDM) were provided by Shanghai Macklin Co., Ltd., Shanghai, China. Diglycidyl ether of bisphenol A type epoxy resin (commercial name: E 51) was purchased from Xingchen Epoxy Resins Factory (Nantong, China). Melamine (MEL) was obtained from Shanghai Lingfeng Co., Ltd., Shanghai, China. Formaldehyde, *N*,*N*-dimethylformamide (DMF), anhydrous copper acetate (Cu(OAc)_2_), trimesic acid (H_3_BTC,C_9_H_6_O_6_), glacial acetic acid (CH_3_COOH), diethyl ether, ethanol, and phenol were purchased from Nanjing Chemical Reagent Co., Ltd., Nanjing, China.

### 3.2. Phenolation of Alkaline Lignin(Ph-Lignin)

We prepared 2 mol/L sulfuric acid (H_2_SO_4_) solution, took 80 mL with a three-neck flask and set the temperature at 80 °C. Then we added 20 g of lignin (Al-lignin) and stirred at condensation reflux for 1.5 h. Then we slowly added 18 g of phenol, dropwise, during the temperature rise to 95 °C and stirred at condensation reflux for 1.5 h at 95 °C. After the reaction was completed, it was cooled to room temperature, the solid was rinsed three times using ether, and then dried at 70 °C for 12 h.

### 3.3. Preparation of MEL and DOPO Mixture(MEL-DOPO)

We combined 280 mL of ethanol aqueous solution (70 wt.%) and 12.96 g of DOPO (0.06 mol), and the mixture was agitated for 30 minutes to thoroughly dissolve the DOPO. When the temperature reached 70 °C, 10 g of MEL (0.08 mol) was added and agitated for 6 h. Following completion, the mixture was cooled to ambient temperature, three times washed with ethanol, and then dried for 12 h in a vacuum oven at 70 °C.

### 3.4. Preparation of PN-Lignin

We added 4 g of phenolized lignin (Ph-lignin) and 20 g of MEL-DOPO to a 50 mL single-mouth flask, then 300 mL of *N*,*N* dimethylformamide (DMF) solution was added and the temperature was set to 75 °C. We added 7.2 g (0.024 mol) of formaldehyde to react for 3 h. After being cooled to room temperature, the solid was repeatedly washed several times by adding an appropriate amount of distilled water, and then the solid was dried in a drying oven at 70 °C for 24 h.

### 3.5. Preparation of PN-Lignin@HKUST-1

First of all, exactly 1.467 g of CMC (6.057 mol) with 2 g of PN-lignin was added into 70 mL of deionized water and mechanically stirred at room temperature for 1 h. Secondly, 1.326 g of Cu(OAc)_2_ (6.642 mmol) dissolved in 30 mL of deionized water and 1 mL of HAc were added dropwise and stirred for 5 min. Thirdly, 1.155 g of H3BTC (5.496 mmol) dissolved in 15 mL ethanol was sequentially added and the mixture was continuously stirred in a closed system for 4 h. Finally, the end product was obtained by centrifugation and dried in the vacuum oven at 70 °C for 24 h.

### 3.6. Preparation of Flame-Retardant EP Composites

A traditional curing method was applied to fabricate epoxy composites with some modifications. Typically, 3 g of PN-lignin@HKUST-1 and 15 g of EP were mechanically stirred at 1000 rpm for 1.5 h to obtain a uniform mixture system. Next, 3 g of curing agent DDM (mass ratio, DDM/EP = 1:5) were added and stirred for 1.5 h. The uniformly-mixed thermoset was poured into the mould and cured at 100 °C for 2 h after bubbles were removed in the vacuum oven. For further curing, the temperature was raised to 150 °C for 2 h. To evaluate the effect of modification, the neat EP thermoset and EP composites containing PN-lignin (referred to as EP/PN-lignin) were also prepared according to the similar procedure.

### 3.7. Measurements

Using the DTG-60AH, the thermogravimetric analysis (TGA) was achieved (SHIMADZU, Kyoto, Japan). The temperature range was from 30 °C to 900 °C, the sample value range was 5–10 mg, nitrogen was used as the protective gas, and the gas flow rate was set at 20 mL/min.

Al-lignin was tested at 8 °C/min, 12 °C/min, 16 °C/min and 20 °C/min. The ramp rates were set at 5 °C/min, 10 °C/min, 15 °C/min and 20 °C/min for EP/15% Flignin@HKUST-1.

### 3.8. Kinetic Modeling

The non-isothermal constant-rate-of-warming approach is frequently employed in thermal analysis to analyze the kinetic behavior of pyrolysis of solid-state reactants. As demonstrated in Equation (3), the thermal decomposition rate for the pyrolytic behavior of solid-state reactants can be stated as a function of temperature *k*(*T*) and conversion *f*(*α*):(3)$$\frac{d\alpha }{dt}=\beta \frac{d\alpha }{dT}=k\left(T\right)f\left(\alpha \right)$$
where

*α* is the conversion of the sample (%)

*β* is the constant temperature rise rate (°C/min)

*T* is the reaction temperature (Kelvin, K)

where the conversion α can be obtained by Equation (4):(4)$$\alpha =\frac{M_{0}-M_{T}}{M_{0}-M_{\infty }}$$
where

*M*_0_ is the initial mass of the sample (%)

*M_T_* is the mass of the sample at the decomposition temperature (%)

*M*_∞_ is the mass of the sample after decomposition (%)

The function *k*(*T*) can generally be obtained from the Arrhenius equation:(5)$$k\left(T\right)=A\times \exp\left(-\frac{E}{RT}\right)$$
where

*A* is the prefactor

*E* is the apparent activation energy (KJ/mol)

*R* is the gas constant (8.3145 J/mol/K)

To generate Equation (7) by integrating the left and right sides of the equation from 0 to and *T*_0_ to *T*, respectively, and designating the integral kinetic mode function as *G*(*α*), Equation (5) can be substituted into 1 and distorted to produce Equation (6). The initial temperature *T_0_* of the reaction is low, and consequently, the reaction rate is very small.
(6)$$\frac{d\alpha }{f\left(\alpha \right)}=\frac{A}{\beta }\exp\left(-\frac{E}{RT}\right)dT$$
(7)$$G\left(\alpha \right)=\int_{0}^{\alpha }\frac{d\alpha }{f\left(\alpha \right)}=\int_{0}^{T}\frac{A}{\beta }\exp\left(-\frac{E}{RT}\right)dT$$

If it is stipulated that:$$u=\frac{E}{RT},\;p\left(u\right)=\int_{\infty }^{u}\frac{-e^{u}}{u^{2}}du$$

The class I kinetic equations for thermal analysis can be obtained (Equation (8)).
(8)$$G\left(\alpha \right)=\int_{0}^{\alpha }\frac{d\alpha }{f\left(\alpha \right)}=\int_{0}^{T}\frac{A}{\beta }\exp\left(-\frac{E}{RT}\right)dT=\frac{AE}{\beta R}\int_{\infty }^{u}\frac{-e^{u}}{u^{2}}du=\frac{AE}{\beta R}P\left(u\right)$$

If *P*(*u*) = *PFK*(*u*) = *e − u/u^*2*^* or take the Doyle approximation, i.e., *P*(*u*) = *P_D_*(*u*) = 0.00484*e* − 1.0516u, the Kissinger equation (Equation (9)) and the Ozawa equation (Equation (10)) can be obtained by deformation when substituting Equation (8), respectively.
(9)$$\ln\left(\frac{\beta }{T}\right)=\ln\left[\frac{AR}{EG\left(\alpha \right)}\right]-\frac{E}{RT}$$
(10)$$\ln\left(\beta \right)=\ln\left[\frac{0.00484AE}{RG\left(\alpha \right)}\right]-\frac{1.0516E}{RT}$$

If the function *P*(*u*) takes the Senum-Yang approximation (*P*(*u*) = *P_SY_*(*u*) = [(*e − u/u^*2*^*) × *Q*_4_(*u*)]), where
(11)$$Q_{4}\left(u\right)=\frac{u^{3}+18u^{2}+88u+96}{u^{4}+20u^{3}+120u^{2}+240u+120}$$
(12)$$H(u)=\frac{P_{SY}\left(u\right)}{P_{D}\left(u\right)}=\frac{e^{-u}Q_{4}\left(u\right)}{0.00484u^{2}e^{-1.0516u}}$$

Immediately combining the aforementioned equations yields the Kissinger-Senum-Yang-iteration equation (Equation (13)) and the Ozawa-Senum-Yang-iteration equation (Equation (14)).
(13)$$\ln\left(\frac{\beta }{Q_{4}\left(u\right)T^{2}}\right)=\ln\left[\frac{AR}{EG\left(\alpha \right)}\right]-\frac{E}{RT}$$
(14)$$\ln\left(\frac{\beta }{H(u)}\right)=\ln\left[\frac{0.00484AE}{RG\left(\alpha \right)}\right]-\frac{1.0516E}{RT}$$

The Kissinger-iterative equation and the Ozawa-iterative equation, both of which take into account how *H*(*u*) and *Q*(*u*) fluctuate slowly with u without being constrained by the range of u, are both members of the equal conversion integration method. To locate *E*, the iterative technique is utilized. To find E, first set *H*(*u*) = 1 or *Q*_4_(*u*) = 1. The starting value of *E* is then determined from the slope using the least-squares approach and the linear relationship between ln(1/*T^*2*^*) − 1/*T* and ln(−1/*T*) using the least-squares method. The new value of E is then calculated from the slope by substituting the initial value into *H*(*u*) and *Q*(*u*) using the linear connection with ln(1/*H*(*u*) − 1/*T* and ln[1/*Q*(*u*)*T^*2*^*] − 1/*T*, respectively. Another corrected value can be derived by iterating once more using this corrected value of *E* as the initial value. A more logical E-value fulfilling less than 0.1 kJ/mol is thus reached after a number of repetitions.

## 4. Conclusions

In this study, Al-lignin and EP/15% F-lignin@HKUST-1 samples were analyzed kinetically using the Kissinger-SY, Ozawa-SY, Lee-Beck approximation, and Gorbatchev approximation iteration methods. The differential-equation method and the universal-integral methods were used to calculate the two samples’ most likely mechanism functions.
Four kinetic methods were used to determine the activation energies of the Al-lignin and EP/15% F-lignin@HKUST-1 samples. The four procedures were reasonable since they produced outcomes that were almost identical. Al-lignin pyrolysis has a wide range of apparent activation energies and can be separated into three stages. The complexity of the pyrolysis process is reflected in the multiple increases and drops in the apparent activation energy of al-lignin pyrolysis. The change of apparent activation energy during pyrolysis of EP/15% F-lignin@HKUST-1 samples can also be divided into three stages, but different from lignin, the apparent activation energy during pyrolysis of these two samples experienced three stages: rapid increase, slow change, and rapid rise, reflecting that their pyrolysis-rate curves only have a single obvious rate peak.Using universal integral and differential equation approaches, the most likely mechanism functions for the pyrolysis of Al-lignin and EP/15% F-lignin@HKUST-1 were identified as AE6, AE4, and AE3. The regression-intercept of the Kissinger-SY and Ozawa-SY iterative approaches were used to get the relevant preexponential component *lnA*, and the related apparent activation energy Eα had a strong linear relationship.

## Figures and Tables

**Figure 1 molecules-28-02699-f001:**
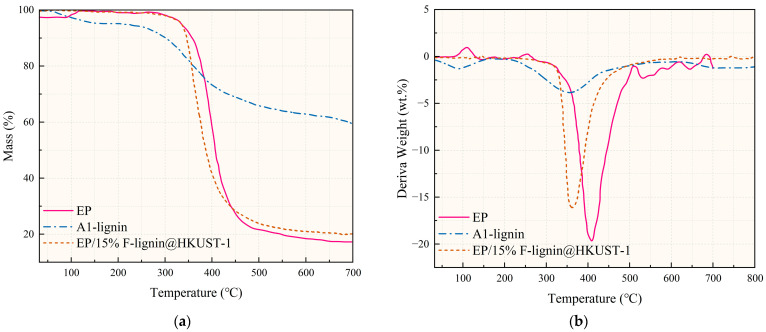
TG (**a**) and DTG (**b**) curves of pure EP and EP composites under N_2_ atmosphere.

**Figure 2 molecules-28-02699-f002:**
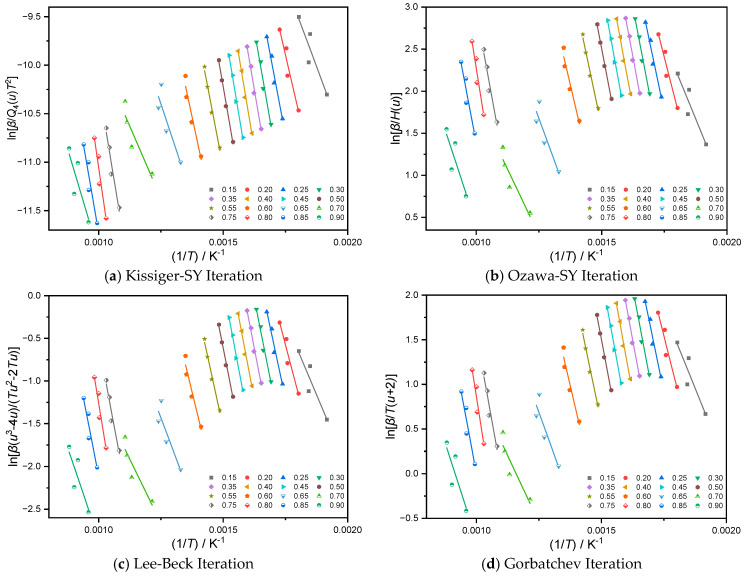
Linear regression fitting diagram of the equal conversion method of Al-lignin, based on four temperature approximations.

**Figure 3 molecules-28-02699-f003:**
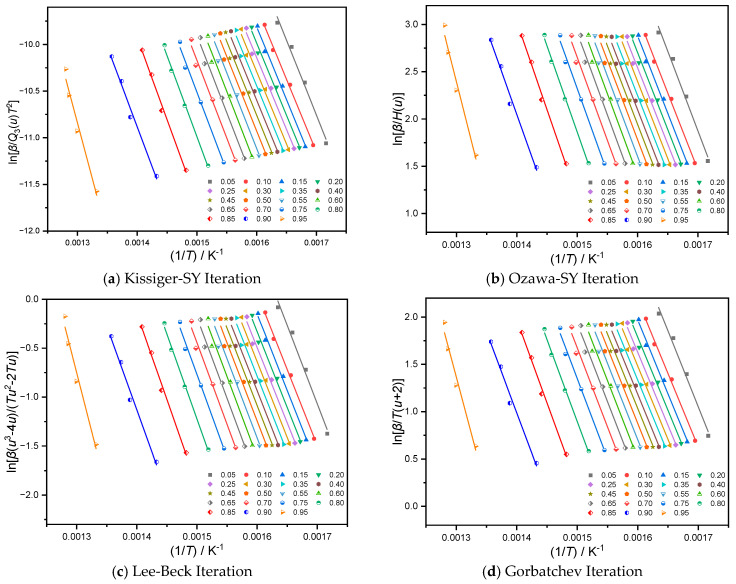
Linear regression fitting diagram of equal conversion method of EP/15% F-Lignin@HKUST-1 based on four temperature approximation.

**Figure 4 molecules-28-02699-f004:**
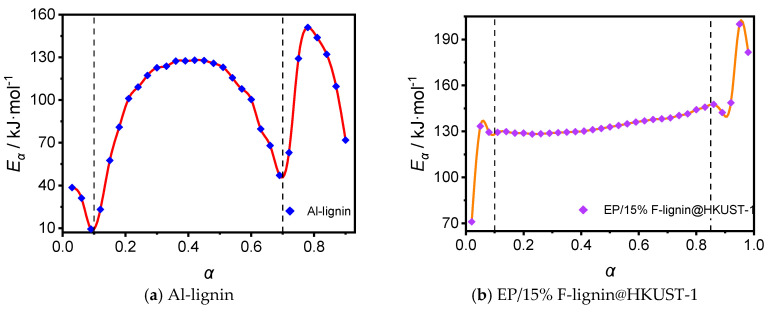
Relationship between activation energy and conversion of Al-lignin (**a**) and EP/15% F-lignin@HKUST-1.

**Figure 5 molecules-28-02699-f005:**
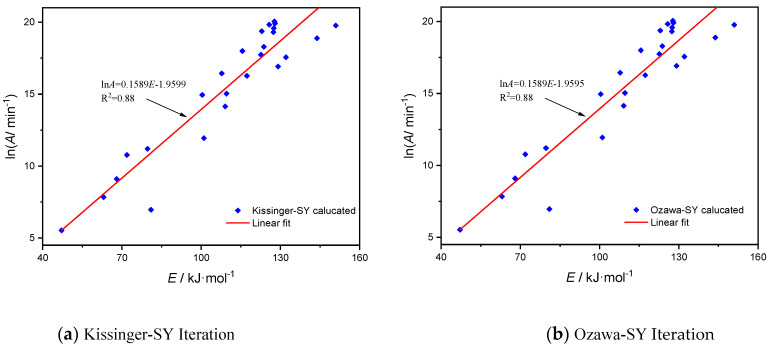
Linear correlation between the activation energy of Al-lignin and the preinhetial factor based on two iterative methods.

**Figure 6 molecules-28-02699-f006:**
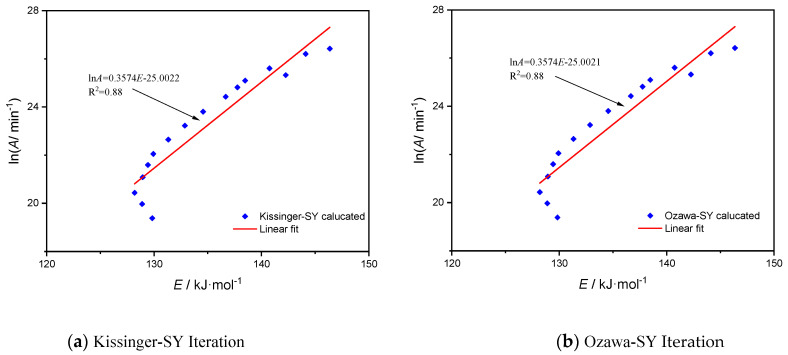
Linear correlation between the activation energy of EP/15% F-lignin@HKUST-1 and the preinhetial factor based on two iterative methods.

**Table 1 molecules-28-02699-t001:** TG data of EP, Al-lignin and EP composites under N_2_ atmosphere.

Samples	T_d5%_ (°C)	T_dmax_ (°C)	Residues (wt.%)
EP	333.7	405.3	16.21
Al-lignin	105.5	333.3	41.57
EP/15%F-lignin@HKUST-1	334.1	360.7	20.14

**Table 2 molecules-28-02699-t002:** Activation energy of Lignin at different conversions.

α	Kissiger-SY Iteration		Ozawa-SY Iteration		Lee-Beck Iteration		Gorbatchev Iteration	
	*E*/(KJ/mol)	R^2^	*E*/(KJ/mol)	R^2^	*E*/(KJ/mol)	R^2^	*E*/(KJ/mol)	R^2^
0.15	57.61	0.92	57.61	0.93	57.67	0.92	57.67	0.92
0.20	92.19	0.96	92.19	0.97	92.22	0.96	92.22	0.96
0.25	112.61	0.98	112.62	0.99	112.64	0.98	112.64	0.98
0.30	122.64	0.99	122.64	0.99	122.67	0.99.	122.67	0.99
0.35	126.60	0.99	126.60	0.99	126.62	0.99	126.62	0.99
0.40	127.89	0.99	127.89	0.99	127.92	0.99	127.92	0.99
0.45	127.73	0.99	127.73	0.99	127.76	0.99	127.76	0.99
0.50	123.28	0.99	123.28	0.99	123..31	0.99	123.31	0.99
0.55	111.54	0.99	111.54	0.99	111.58	0.99	111.58	0.99
0.60	100.41	0.97	100.42	0.97	100.47	0.97	100.47	0.97
0.65	64.24	0.89	64.24	0.90	64.37	0.89	64.37	0.89
0.70	49.34	0.91	49.34	0.92	49.56	0.92	49.56	0.92
0.75	129.12	0.97	129.12	0.97	129.19	0.97	129.19	0.97
0.80	148.40	0.96	148.41	0.97	148.47	0.96	148.47	0.96
0.85	124.02	0.95	124.02	0.95	124.12	0.95	124.12	0.94
0.90	71.87	0.84	71.87	0.85	72. 10	0.84	72.10	0.84

**Table 3 molecules-28-02699-t003:** Activation energy of EP/15% F-Lignin@HKUST-1 at different conversions.

α	Kissiger-SY Iteration		Ozawa-SY Iteration		Lee-Beck Iteration		Gorbatchev Iteration	
	*E*/(KJ/mol)	R^2^	*E*/(KJ/mol)	R^2^	*E*/(KJ/mol)	R^2^	*E*/(KJ/mol)	R^2^
0.15	133.40	0.95	133.40	0.94	133.42	0.93	133.42	0.95
0.20	129.26	0.96	129.26	0.96	129.29	0.96	129.29	0.95
0.25	129.84	0.97	129.84	0.97	128.91	0.98	129.86	0.97
0.30	128.89	0.98	128.89	0.99	128.22	0.99	128.91	0.98
0.35	128.19	0.99	128.19	0.98	128.97	0.99	128.22	0.99
0.40	128.94	0.99	128.94	0.99	129.46	0.99	128.97	0.99
0.45	129.44	0.99	129.44	0.99	1 29.95	0.99	129.46	0.99
0.50	129.93	0.99	129.93	0.99	131.36	0.99	129.95	0.99
0.55	131.33	0.99	131.33	0.99	132.90	0.99	131.36	0.99
0.60	132.87	0.98	132.87	0.99	134.60	0.99	132.90	0.96
0.65	134.57	0.98	134.57	0.97	136.70	0.98	134.60	0.97
0.70	136.67	0.95	136.67	0.94	137.78	0.96	136.70	0.95
0.75	137.76	0.95	137.76	0.92	138.50	0.96	137.78	0.95
0.80	138.48	0.96	138.48	0.91	140.78	0.94	138.50	0.94
0.85	140.75	0.95	140.75	0.92	144.14	0.95	140.78	0.93
0.90	144.11	0.95	144.1 1	0.91	146.39	0.94	144.14	0.94

**Table 4 molecules-28-02699-t004:** Kinetic parameters of universal integral equations and differential equations based on the Avrami-Erofeev equation at different temperature rates of Al-lignin.

Mechanism Function	β /K∙min^−1^	Universal Integration			Differential Equation		
		E/ KJ∙mol^−1^	Ln (A/min^−1^)	R^2^	E/ KJ∙mol^−1^	Ln (A/min^−1^)	R^2^
AE_4_	8	65.27	5.97	0.97	71.62	6.79	0.97
	12	74.16	7.32	0.97	75.81	7.38	0.96
	16	71.87	7.19	0.98	76.89	7.77	0.97
	20	75.48	7.77	0.96	72.49	7.22	0.95
AE_5_	8	84.26	8.9	0.98	90.61	9.73	0.98
	12	95.36	10.49	0.98	97.04	10.55	0.97
	16	92.50	10.25	0.98	97.54	10.84	0.98
	20	97.04	10.93	0.97	94.10	10.37	0.96
AE_6_	8	103.25	11.82	0.98	109.6	12.67	0.98
	12	116.57	13.65	0.98	118.27	13.72	0.97
	16	113.13	13.31	0.98	118.19	13.91	0.98
	20	118.61	14.08	0.97	115.7	13.53	0.96

**Table 5 molecules-28-02699-t005:** Kinetic parameters of universal integral equations and differential equations based on the Avrami-Erofeev equation at different temperature rates of EP/15% F-lignin@HKUST-1.

Mechanism Function	β /K∙min^−1^	Universal Integration			Differential Equation		
		E/ KJ∙mol^−1^	Ln (A/min^−1^)	R^2^	E/ KJ∙mol^−1^	Ln (A/min^−1^)	R^2^
AE_4_	5	73.96	8.97	0.92	72.47	10.37	0.90
	10	95.41	11.84	0.93	86.88	12.60	0.91
	15	81.44	9.99	0.95	80.07	12.08	0.94
	20	112.18	14.86	0.94	101.86	15.73	0.92
AE_5_	5	116.85	16.31	0.93	115.22	17.68	0.92
	10	149.73	20.74	0.94	140.93	21.42	0.93
	15	127.66	17.57	0.96	126.24	19.64	0.95
	20	174.21	24.81	0.95	163.81	25.63	0.94
AE_6_	5	159.74	23.66	0.93	158.04	25.01	0.93
	10	204.05	29.63	0.94	195.08	30.27	0.94
	15	173.89	25 15	0.96	172 43	27 22	0.96
	20	236.24	34.75	0.95	225.78	35.55	0.94

**Table 6 molecules-28-02699-t006:** Activation energy and preintential factors of Al-lignin at different conversions under Kissinger-SY iteration and Ozawa-SY iteration method.

α	Kissinger-SY Iteration		Ozawa-SY Iteration	
	*E*/KJ∙mol^−1^	*lnA*/min^−1^	*E*/KJ∙mol^−1^	*lnA*/min^−1^
0.15	57.61	0.92	57.61	0.92
0.20	92.19	9.83	92.19.	9.83
0.25	112.61	15.02	112.62	15.02
0.30	122.64	17.74	122.64	17.74
0.35	126.6	19.07	126.6	19.07
0.40	127.89	19.75	127.89	19.75
0.45	127.73	20.06	127.73	20.06
0.50	123.28	19.43	123.28	19.43
0.55	111.54	17.22	111.54	17.22
0.60	100.41	14.95	100.42	14.95
0.65	64.24	8.53	64.24	8.53
0.70	49.34	5.84	49.34	5.84
0.75	129.12	16.92	129.12	16.92
0.80	148.4	19.43	148.41	19.43
0.85	124.02	16.63	124.02	16.63
0.90	71.87	10.80	71.87	10.80

**Table 7 molecules-28-02699-t007:** Activation energy and preintential factors of EP/15% F-lignin@HKUST-1 at different conversions under Kissinger-SY iteration and Ozawa-SY iteration method.

α	Kissinger-SY Iteration		Ozawa-SY Iteration	
	*E*/KJ∙mol^−1^	*lnA*/min^−1^	*E*/KJ∙mol^−1^	*lnA*/min^−1^
0.15	129.84	19.38	129.84	19.38
0.20	128.89	19.97	128.89	19.97
0.25	128.19	20.44	128.19	20.44
0.30	128.94	21.08	128.94	21.08
0.35	129.44	21.59	129.44	21.59
0.40	129.93	22.05	129.93	22.05
0.45	131.33	22.64	131.33	22.64
0.50	132.87	23.22	132.87	23.22
0.55	134.57	23.80	134.57	23.80
0.60	136.67	24.42	136.67	24.42
0.65	137.76.	24.81	137.76.	24.81
0.70	13848	25.09	13848	25.09
0.75	140.75	25.60	140.75	25.60
0.80	144.11	26.20	144.11	26.20
0.85	146.36	26.42	146.36	26.42
0.90	142.26	25.32	142.26	25.32

## Data Availability

Not applicable.

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
