# Peer review of "Pyrolysis Kinetics of Lignin-Based Flame Retardants Containing MOFs Structure for Epoxy Resins"

_molecules, 2023, doi:10.3390/molecules28062699_

Round 1

Reviewer 1 Report

The generally accepted style of writing formulas of chemical compounds is not respected (L. 98, 99, 103, 123,194, 210, 211)

Section 2.7 lacks timing

The style of writing the same symbols, and their indices in formulas and text is different

The designation of the regression coefficient in the text and tables is different

a in formula (2) and later in the text is not the conversion rate. It's just the conversion itself.

The text and table 2 do not indicate to which temperature the reduced residual value refers (char residual)

The paper compares the behavior of a EP, Al-lignin and a EP with modified lignin EP/F-lignin@HKUST-1. However, the meaning of lignin modification is not clear, since there is no comparison of the behavior of Al-lignin with modified lignin F-lignin@HKUST-1, as well as a comparison of a EP/Al-lignin with a EP/F-lignin@HKUST-1.

Author Response

Response:

Thank you very much for your comments. We take the liberty to address the concerns in the point-to-point style as below.

Reviewer #1:

  1. The generally accepted style of writing formulas of chemical compounds is not respected (L. 98, 99, 103, 123,194, 210, 211).

Response 1: We thank the reviewer for this valuable comment. We have made corrections.

  1. Section 2.7 lacks timing.

Response 2: We thank the reviewer for this valuable comment. In this paper, the thermogravimetric analysis data of samples at different heating rates in a specified temperature range are investigated. At a certain heating rate, when the temperature range is determined, the heating time is also determined.

  1. The designation of the regression coefficient in the text and tables is different.

Response 3: We thank the reviewer for this valuable comment. We have made corrections. ‘With the increased conversion (0.20 < α < 0.80), the regression coefficients' R2 values ranged from 0.95 to 0.99.’

  1. a in formula (2) and later in the text is not the conversion rate. It's just the conversion itself.

Response 4: We thank the reviewer for this valuable comment. We have made corrections.

  1. The text and table 2 do not indicate to which temperature the reduced residual value refers (char residual)

Response 5: We thank the reviewer for this valuable comment. We have made corrections, e.g., (from room temperature to 700 °C).

  1. The paper compares the behavior of a EP, Al-lignin and a EP with modified lignin EP/F-lignin@HKUST-1. However, the meaning of lignin modification is not clear, since there is no comparison of the behavior of Al-lignin with modified lignin F-lignin@HKUST-1, as well as a comparison of a EP/Al-lignin with a EP/F-lignin@HKUST-1.

Response 6: We thank the reviewer for this valuable comment. And we are preparing our next manuscript to Molecules about difference between EP, lignin, EP with modified lignin, and EP/F-lignin supported on MOFs. We appreciate your important advice.

Reviewer 2 Report

In this work, lignin was modified by phenol and amination, and then 9,10-dihydro-9-oxa-10-phosphophenanthrene-10-oxide (DOPO) was introduced to prepare a lignin-based flame retardant (Lignin-N-DOPO) containing nitrogen and phosphorus flame retardant elements. On this basis, the MOFs structures HKUST-1 was successfully introduced to prepare HKUST-1/lignin-based flame retardant (F-lignin@HKUST-1). The thermal properties of epoxy resin composites were studied by thermogravimetric analysis and the activation energy was measured by various kinetic methods.

However, the following issues need to be resolved before being published.

1) What does PP, PLA and PU mean? Abbreviations used should be explained where they are first used.

2) In the introduction, MOFs materials are used in flame retardant materials due to their structural characteristics, their good adsorption effect on smoke generation and their good catalytic effect on CO oxidation. Authors can refer to several relevant references.

3) Section 2.2 Preparation of Ph-lignin. The authors repeatedly state that the products were washed by filtration using 500 ml of
diethyl ether. Is the 500 ml mentioned here a specific value, and does the use of too much or too little diethyl ether affect the products? If not, it is suggested to use the word "moderate" instead.

The same question applies to subsection 2.4 Preparation of PN-lignin

4) In Figure 1, The horizontal coordinates of the TG(A) and DTG(B) curves should be the same.

5) The quality of the English language can be improved. Thoroughly check the research article's punctuation marks, grammar, and spelling errors.

Author Response

Comments of Reviewer 2 and our replies

Response:

Thank you very much for your comments. We take the liberty to address the concerns in the point-to-point style as below.

Reviewer #2:

In this work, lignin was modified by phenol and amination, and then 9,10-dihydro-9-oxa-10-phosphophenanthrene-10-oxide (DOPO) was introduced to prepare a lignin-based flame retardant (Lignin-N-DOPO) containing nitrogen and phosphorus flame retardant elements. On this basis, the MOFs structures HKUST-1 was successfully introduced to prepare HKUST-1/lignin-based flame retardant (F-lignin@HKUST-1). The thermal properties of epoxy resin composites were studied by thermogravimetric analysis and the activation energy was measured by various kinetic methods.

However, the following issues need to be resolved before being published.

  1. What does PP, PLA and PU mean? Abbreviations used should be explained where they are first used.

Response 1: We thank the reviewer for this valuable comment. PP means Polypropylene, PLA means polylactic acid, PU means polyurethane.

  1. In the introduction, MOFs materials are used in flame retardant materials due to their structural characteristics, their good adsorption effect on smoke generation and their good catalytic effect on CO oxidation. Authors can refer to several relevant references.

10.1016/j.cej.2019.122777

10.1016/j.jhazmat.2018.09.086

10.1021/acs.iecr.6b04920

10.1016/j.compscitech.2017.08.032

Response 2: We thank the reviewer for this valuable comment. We have added this literature to the introductory chapter.

Zhang et al. used MOFs as precursors to link two different LDHs into composite nanomaterials, and the specimens passed the V-0 rating in the UL-94 test.

Xu et al. prepared functionalized reduced graphene oxide (RGO) with Co-ZIF (zeolite imidazolium framework-67) adsorbed borate ions (ZIF-67/RGO-B). The peak heat release rate (PHRR) total heat release rate (THR) and maximum smoke concentration (Ds,max) of the composites with ZIF-67/RGO-B doping of 2 wt.% were reduced by 65.1%, 41.1% and 66.0%, respectively, compared to the pure EP.

Hou et al. successfully synthesized iron- and cobalt-based metal-organic frameworks (MOFs) and incorporated the resulting MOFs into polystyrene (PS) as flame retardants. Thermogravimetric and cone calorimetric analyses showed that the MOFs had good flame retardant effects.

Hou et al. synthesized a Co-based metal organic backbone (P-MOF) with a phosphorus-based structure by a facile hydrothermal reaction and added it to an epoxy resin (EP) to enhance its flame retardancy. The test results from conical calorimeter and steady-state tube furnace showed a significant reduction in total flue gas yield and total CO.

  1. Section 2.2 Preparation of Ph-lignin. The authors repeatedly state that the products were washed by filtration using 500 ml of diethyl ether. Is the 500 ml mentioned here a specific value, and does the use of too much or too little diethyl ether affect the products? If not, it is suggested to use the word "moderate" instead.

The same question applies to subsection 2.4 Preparation of PN-lignin.

Response 3: We thank the reviewer for this valuable comment. We have reworked the description.

Prepare 2 mol/L sulfuric acid (H2SO4) solution, take 80 mL with a three-neck flask and set the temperature at 80 °C. Add 20 g of lignin (Al-lignin) and stir at condensation reflux for 1.5 h. Then slowly add 18 g of phenol dropwise during the temperature rise to 95 °C and stir at condensation reflux for 1.5 h at 95 °C. After the reaction was completed, it was cooled to room temperature, the solid was rinsed 3 times using ether, and then dried at 70 °C for 12 h.

4 g of phenolized lignin (Ph-lignin) and 20 g of MEL-DOPO were added to a 50 mL single mouth flask, then 300 mL of N,N dimethylformamide (DMF) solution was added and the temperature was set to 75 °C. 7.2 g (0.024 mol) of formaldehyde was added to react for 3 h. After being cooled to room temperature, the solid was repeatedly washed several times by adding appropriate amount of distilled water, and then the solid was dried in a drying oven at 70 °C for 24 h.

  1. In Figure 1, The horizontal coordinates of the TG(A) and DTG(B) curves should be the same.

Response 4: We thank the reviewer for this valuable comment. We have redrawn Figure 1.

  1. The quality of the English language can be improved. Thoroughly check the research article's punctuation marks, grammar, and spelling errors.

Response 4: We thank the reviewer for this valuable comment. We have revised the manuscript.

------------------------------------------------------

Sincerely yours,

Tianyu Yao 1,†, Ruohan Yang 1,†, Cong Sun 1, Yuzhu Lin 1, Ruoqi Liu 1, Hongyu Yang 1, Jiajia Chen 1,*, Xiaoli Gu 1,*

Round 2

Reviewer 1 Report

The style of writing the same symbols, and their indices in formulas and text is different (M0 and M0 etc)

The paper compares the behavior of a EP, Al-lignin and a EP with modified lignin EP/F-lignin@HKUST-1. However, the meaning of lignin modification is not clear, since there is no comparison of the behavior of Al-lignin with modified lignin F-lignin@HKUST-1, as well as a comparison of a EP/Al-lignin with a EP/F-lignin@HKUST-1.

Author Response

The purpose of this manuscript is to investigate the modification effect of epoxy (EP) resin, a conventional thermosetting plastic. The most common and effective method is adding flame retardants in the preparation of epoxy resin. Therefore, we introduce a green and convenient method by blending with as-prepared flame retardant (referred to as EP/F-lignin@HKUST-1). So we did not compare the difference between lignin and modified lignin.

We really thank the reviewer for valuable comment. And we are preparing our next manuscript about flame retardancy with or without addition of modified lignin and modified lignin@MOFs. THANKS!